# Structural insight into the intraflagellar transport complex IFT-A and its assembly in the anterograde IFT train

Yuanyuan Ma [1,2,5], Jun He[1,2,5], Shaobai Li[1,2], Deqiang Yao[3], Chenhui Huang [1,2], Jian Wu [1,2] ✉ & Ming Lei [1,2,4] ✉

Intraflagellar transport (IFT) trains, the polymers composed of two multi-subunit complexes, IFT-A and IFT-B, carry out bidirectional intracellular transport in cilia, vital for cilia biogenesis and signaling. IFT-A plays crucial roles in the ciliary import of membrane proteins and the retrograde cargo trafficking. However, the molecular architecture of IFT-A and the assembly mechanism of the IFT-A into the IFT trains in vivo remains elusive. Here, we report the cryo-electron microscopic structures of the IFT-A complex from protozoa *Tetrahymena thermophila*. We find that IFT-A complexes present two distinct, elongated and folded states. Remarkably, comparison with the in situ cryo-electron tomography structure of the anterograde IFT train unveils a series of adjustments of the flexible arms in apo IFT-A when incorporated into the anterograde train. Our results provide an atomic-resolution model for the IFT-A complex and valuable insights into the assembly mechanism of ante-rograde IFT trains.

Cilia are highly conserved organelles that protrude from the cell surface and serve a wide variety of roles[1]. Dysfunction of cilia leads to numerous human diseases called ciliopathies that are associated with vision impairment, kidney dysfunction and male infertility[2]. The intraflagellar transport (IFT) machinery is essential for the assembly and maintenance of the cilium as well as signal transduction by delivery of cargoes into and out of cilia[3,4].

Each IFT train is a polymer of two large complexes IFT-A and IFT-B with molecular weight of ~0.8 MDa and 1 MDa, respectively[5]. The IFT-B complex comprises of a 10-subunit core subcomplex IFT-B1 and a 6-subunit peripheral subcomplex IFT-B2, mediating the anterograde trafficking (toward the tip) of ciliary proteins driven by the micro-tubule motor kinesin-2[6–8]. The IFT-A complex is composed of six sub-units (IFT144, IFT140, IFT122, IFT121, IFT139 and IFT43), mediating both the retrograde trafficking (toward the base) driven by the dynein-1b and the ciliary import of various membrane proteins across the ciliary gate[9–14]. The in situ cryo-electron tomography (cryo-ET) study of

the anterograde train in *Chlamydomonas reinhardtii* illuminates that the anterograde train is densely packed with IFT-B as the backbone lying between IFT-A and the axoneme, and IFT-A located directly underneath the ciliary membrane[15]. After reaching the ciliary tip, the anterograde trains are remodeled to completely distinct zigzagged retrograde trains, coordinating multiple events including inactivation of kinesin-2, activation of dynein-1b and release of cargoes[15–20]. Defects or reduced concentration of IFT-A leads to short cilia with decreased velocities of retrograde transport and the accumulation of IFT-B at the ciliary tip[21–25]. Mutations in IFT-A genes caused a plethora of cilio-pathies including skeleton, kidney and eye associated diseases[2,26]. Lack of high-resolution IFT-A structures greatly hinders our understanding of the assembly mechanism of IFT-A into the IFT trains and of the molecular basis of ciliary trafficking and ciliopathies.

Recently, several studies reported high-resolution cryo-electron microscopy (cryo-EM) IFT-A structures as well as a 20.7-Å in-situ cryo-ET model of the anterograde IFT train[27–30]. Here, we purified the IFT-A

[1]Ninth People's Hospital, Shanghai Jiao Tong University School of Medicine, Shanghai 200011, China. [2]Shanghai Institute of Precision Medicine, Shanghai 200125, China. [3]Renji Hospital, Shanghai Jiao Tong University School of Medicine, Shanghai 200032, China. [4]State Key Laboratory of Oncogenes and Related Genes, Shanghai Jiao Tong University School of Medicine, Shanghai 200025, China. [5]These authors contributed equally: Yuanyuan Ma, Jun He. ✉e-mail: wujian@shsmu.edu.cn; leim@shsmu.edu.cn

complex from protozoa *Tetrahymena thermophila* and determined the cryo-EM structures of the IFT-A complex and its subassemblies at a resolution range of 3.6 to 8.8 Å. Analysis of these structures reveal that the apo IFT-A complex is composed of two rigid structural modules, the head and base modules, and presents two distinct–elongated and folded states in solutions, the latter of which was not identified in recent IFT-A structural studies[27–31]. Moreover, we find that IFT-A is incorporated into the anterograde IFT train via a series of adjustments of the flexible arms in the apo IFT-A. Together, our results greatly extend our understanding of the assembly mechanism of IFT trains and provide structural insights into the pathogenesis of disease-related IFT-A variants in human.

## Results

### *Tetrahymena* IFT-A complex adopts a flexible bimodular architecture

To understand the assembly mechanism of the IFT-A complex, we generated a carboxyl-terminally Flag and protein-A tagged IFT122-FZZ strain of *Tetrahymena thermophila*. A two-step affinity purification scheme was employed to enrich the endogenous *Tetrahymena* IFT-A complex. The purified complex was then subjected to SDS-PAGE and mass spectrometry analyses, confirming the presence of all six components of IFT-A (Fig. 1a, b, Supplementary Fig. 1a and Supplementary Data 1). Strikingly, native-PAGE image unveiled two separate smeared bands of IFT-A, suggesting that this highly purified complex adopts two major conformations, a heterogenous slow-migrating state and a more homogenous fast-migrating one (Fig. 1b). Two-dimensional classification of negative staining data unveiled that apo IFT-A particles are highly flexible and adopt either elongated or folded conformations, in accordance with the slow- and fast-migrating states captured by the native-PAGE analysis (Fig. 1b–d).

To determine the cryo-EM structure of *Tetrahymena* IFT-A, we collected 66,729 raw IFT-A particle images in vitrified ice (Supplementary Fig. 1b). Sub-classification allowed us to obtain two distinct classes of particles that yielded the density maps for the elongated and folded IFT-A complex at resolutions of 8.5 Å and 8.8 Å, respectively (Fig. 1e, f and Supplementary Figs. 2,3). Comparative analysis of these two maps revealed that IFT-A is composed of two rigid modules with similar sizes, which hereafter we designate as the head and base modules of IFT-A (Fig. 1e, f). In the elongated complex, the head and base modules adjoin together to form a slim 'S'-shaped architecture, spanning more than 370 Å in the longest dimension (Fig. 1e). In stark contrast, in the folded conformation the head module rotates about 125° as a rigid body from its position in the elongated state to fold back onto the base module, exemplifying the dynamic nature of the apo IFT-A complex (Fig. 1e–g).

### Molecular architecture of the head and base modules of apo IFT-A

To reveal the structure of the IFT-A complex at the atomic level, we employed local focused classification to individually improve the EM density maps of the base and head modules, which enabled us to build the models of both modules by de novo tracing with the aid of AlphaFold-2 prediction (Supplementary Figs. 2–5 and Table 1)[32]. Except for IFT43, all five large components of IFT-A contain a long array of tetratricopeptide repeats (TPRs) that constitute the overall skeleton of the complex (Fig. 1a and Supplementary Fig. 6). In particular, IFT144, IFT140, IFT122 and IFT121, share the same domain organization, from the N- to the C-termini encompassing two tandem WD40 β-propellers (WDs), an array of 12 to 20 TPRs and, except for IFT140, a zinc binding domain (ZBD) (Fig. 1a). These four subunits are distributed into the two modules; the head module consists of IFT140, IFT144 and the C-terminal half of IFT122 (IFT122$_C$, 706–

1246), whereas the base module consists of IFT121, the N-terminus of IFT122 (IFT122$_N$, 1-705) as well as IFT139 and IFT43 (Fig. 2a, b). Intriguingly, the N-terminal WD40 propellers together with the first four TPRs of both IFT140 and IFT144 in the head module are organized by pseudo-two-fold symmetry, in which a unique α-helical insertion between the second and the third TPRs in IFT140 and IFT144 allows TPRs at the junction to lock into a zigzagged configuration in both shape and electrostatic complementary manners (Figs. 1a, 2c and Supplementary Fig. 7a). Similar pseudo-two-fold symmetric configuration is also found in the IFT121-IFT122 heterodimer in the base module, underscoring the conservation of this basic organizational mechanism in both modules of IFT-A (Figs. 1a, 2d and Supplementary Fig. 7b). Consistent with their structural importance, single missense mutations or compound heterozygous variations in the first four TPRs of human IFT121, IFT122, IFT140 and IFT144 cause multiple ciliopathies, including Cranioectodermal dysplasia, Leber congenital amaurosis, and Short-rib thoracic dysplasia[33–36].

Notwithstanding the conserved pseudo-two-fold symmetric organization, the head and base modules show contrasting structural features. In the head module, the ~15-residue linkers between the second WD40 propeller and the first TPR in both IFT140 and IFT144 make a sharp turn such that the propellers of IFT140 and IFT144 point to opposite directions away from the central pseudo-two-fold axis in an elongated conformation (Fig. 2a, c). The C-terminal ZBD of IFT122 attaches to the other side of the IFT140-IFT144 TPR junction to constitute a compact core that is further encircled by the TPR array of IFT140, burying a total of ~2500 Å² solvent exposed surface area (Fig. 2e). From this core extend out the TPR arms of IFT122, IFT140 and IFT144, the most variable segments in the head module (Supplementary Fig. 3). A frameshift mutation at Tyr1077 in human IFT122$_{ZBD}$ (equivalent to Lys1013 in *Tetramymena* IFT122$_{ZBD}$) that would disrupt the head module core causes the severe Beemer-Langer syndrome with early infant death or Cranioectodermal dysplasia[37], highlighting the essential role of IFT122$_{ZBD}$ in organizing the head module core in IFT-A (Fig. 2e).

In contrast to the head-module elongated conformation, the base module adopts a compact configuration (Fig. 2b). IFT121, IFT122 and IFT139 together form an enclosed oval-shaped ring with the N-terminal TPRs of IFT139 (IFT139$_N$, 1-694) extending out in a solenoid conformation (Fig. 2b). The short linkers (~5 residues) between the WD40 propellers and the TPRs in IFT121 and IFT122 constrain the locations of the propellers right atop the IFT121-IFT122 TPR junction to form half of the oval ring (Fig. 2b). This conformation is stabilized by extensive contacts between IFT121 and IFT122, which bury ~3130 Å² solvent exposed surface area of the base module (Fig. 2b, d). On the other side, the ZBD of IFT121 fits into a helical groove formed by the TPRs of IFT139, constituting the other half of the base ring (Fig. 2b). Notably, the base module oval ring is sealed by the very C-terminal residue of IFT139, Lys1334, whose aliphatic sidechain sticks into hydrophobic central cavity of IFT122$_{WD1}$ (Fig. 2f). The highly conserved Lysine or Arginine of IFT139 in all organisms indicates a similar closure mechanism for the base module of IFT-A (Supplementary Fig. 6c).

### IFT43 is a molecular stabilizer of the base module

A salient feature of the base module is that, IFT43, the smallest subunit of the complex, adopts an elongated configuration and traverses the base as a molecular tether to stabilize the ring-shaped architecture (Fig. 2b, f and Supplementary Fig. 8). The N-terminus of IFT43 (IFT43$_N$) travels along the surface of the two WDs of IFT121 with the aromatic sidechains of Trp9-Phe11 and Trp33 deeply embedded into the hydrophobic pockets in IFT121$_{WD1}$ and IFT121$_{WD2}$ respectively, clamping the two propellers in a fixed orientation in the base module (Fig. 2f). Notably, IFT43$_N$ was not observed in recently reported IFT-A

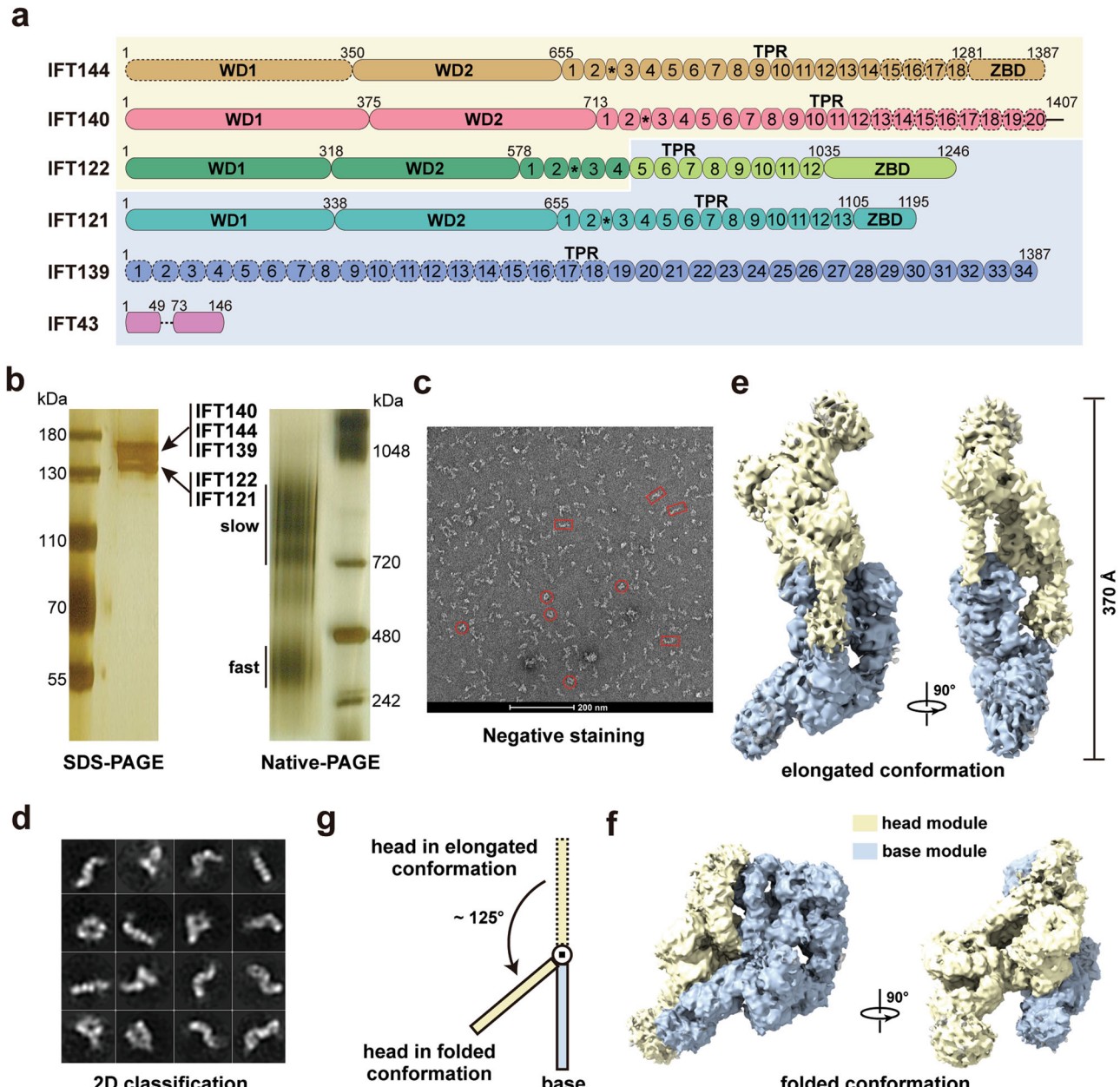

**Fig. 1 | Overall structure of the IFT-A complex in two different conformations. a** Domain organizations of the IFT-A components. IFT144, IFT140, IFT122$_N$, IFT122$_C$, IFT121, IFT139 and IFT43 are colored goldenrod, salmon, green, green yellow, turquoise, blue and violet, respectively. Black asterisks indicate the helices inserted between the second and third TPR repeats in IFT144, IFT140, IFT122, and IFT121. The head and base module components have light yellow and pale blue background colors, respectively. Models of IFT144$_{WD1}$, IFT144$_C$, IFT140$_C$ and IFT139$_N$ built with the aid of AlphaFold-2 are shown in dashed ovals or circles. The middle disordered loop of IFT43 is shown in a dashed line. **b** Silver staining SDS-PAGE gel (left) and native-PAGE gel (right) showing the purity and major conformations of the purified IFT-A complex. Each experiment was repeated three times independently with similar results. **c** A representative negative staining image of the IFT-A complex. Two different conformations of the particles are marked with red rectangle and circle, respectively. **d** 2D class averages of the negative staining images of the IFT-A complex. Cryo-EM density maps of the elongated (**e**) and folded (**f**) IFT-A complex in two orthogonal views. The head and base modules are colored in light yellow and pale blue, respectively. **g** A schematic diagram showing the conformational change of the head module from the elongated to the folded state. Source data are provided as a Source Data file.

structures[27–30]. The C-terminal half of IFT43 (IFT43$_C$) folds into six separate helixes (α1-α6) that meander along the surface of IFT121 and IFT122, burying ~3730 Å² solvent-accessible surface area in the base module (Fig. 2f). In particular, helixes α1-α3 fill into the vacant area between IFT121$_{ZBD}$ and IFT139 whereas helix α4 occupies a deep groove formed by IFT122$_{WD2}$ and IFT121$_{ZBD}$ (Fig. 2f). Loss of IFT43 results in the instability of IFT-A and even loss of flagella[14], underscoring the importance of IFT43's molecular glue function in the assembly of the base module of IFT-A.

## Dynamic conformations of the elongated and folded states of apo IFT-A

To reveal the relationship between the head and base modules in apo IFT-A, we fit the structures of these modules into the elongated and folded IFT-A density maps with manual adjustments to generate the atomic models of the entire IFT-A complex in both states (Fig. 3a). Structural comparison shows that individual head and base modules are almost identical in these two states (Fig. 3a and Supplementary Fig. 9). The most striking difference between the elongated and folded

**Table 1 | Cryo-EM data collection, refinement and validation statistics**

| | IFT-A_Elongated (EMD-34893) | IFT-A_Folded (EMD-34894) | IFT-A_Base-1 (EMD-34895) (PDB 8HMC) | IFT-A_Base-2 (EMD-34896) (PDB 8HMD) | IFT-A_Head-1 (EMD-34897) (PDB 8HME) | IFT-A_Head-2 (EMD-34898) (PDB 8HMF) | IFT-A_Head-3 (EMD-34899) |
|---|---|---|---|---|---|---|---|
| **Data collection and processing** | | | | | | | |
| Magnification | ×81,000 | ×81,000 | ×81,000 | ×81,000 | ×81,000 | ×81,000 | ×81,000 |
| Voltage (kV) | 300 | 300 | 300 | 300 | 300 | 300 | 300 |
| Electron exposure (e⁻/Å²) | 50.0 | 50.0 | 50.0 | 50.0 | 50.0 | 50.0 | 50.0 |
| Defocus range (μm) | 1.5–2.5 | 1.5–2.5 | 1.5–2.5 | 1.5–2.5 | 1.5–2.5 | 1.5–2.5 | 1.5–2.5 |
| Pixel size (Å) | 1.10 | 1.10 | 1.10 | 1.10 | 1.10 | 1.10 | 1.10 |
| Symmetry imposed | C1 | C1 | C1 | C1 | C1 | C1 | C1 |
| Initial particle images (no.) | 1,225,162 | 101,666 | 1,239,500 | 1,239,500 | 1,225,162 | 1,225,162 | 1,225,162 |
| Final particle images (no.) | 35,628 | 14,020 | 538,078 | 86,362 | 386,801 | 199,946 | 41,877 |
| Map resolution (Å) | 8.5 | 8.8 | 3.6 | 4.7 | 4.2 | 4.6 | 6.0 |
| FSC threshold | 0.143 | 0.143 | 0.143 | 0.143 | 0.143 | 0.143 | 0.143 |
| Map resolution range (Å) | 7.0–14.0 | 7.0–12.0 | 3.2–5.0 | 4.0–8.0 | 4.0–6.0 | 4.0–8.0 | 5.5–10.0 |
| **Refinement** | | | | | | | |
| Initial model used (PDB code) | | | n/a | n/a | n/a | n/a | |
| Model resolution (Å) | | | n/a | n/a | n/a | n/a | |
| FSC threshold | | | n/a | n/a | n/a | n/a | |
| Model resolution range (Å) | | | n/a | n/a | n/a | n/a | |
| Map sharpening B factor (Å²) | | | −106 | −120 | −126 | −152 | |
| Model composition | | | | | | | |
| Non-hydrogen atoms | | | 21,797 | 27,372 | 22,169 | 25,330 | |
| Protein residues | | | 2693 | 3387 | 2745 | 3135 | |
| Ligands | | | 2 | 2 | 2 | 2 | |
| B factors (Å²) | | | | | | | |
| Protein | | | 72.06 | 117.41 | 78.33 | 86.15 | |
| Ligand | | | 174.17 | 336.87 | 116.42 | 127.01 | |
| R.m.s. deviations | | | | | | | |
| Bond lengths (Å) | | | 0.011 | 0.003 | 0.002 | 0.002 | |
| Bond angles (°) | | | 1.060 | 0.619 | 0.587 | 0.599 | |
| **Validation** | | | | | | | |
| MolProbity score | | | 1.87 | 1.88 | 1.93 | 1.87 | |
| Clashscore | | | 8.57 | 9.86 | 11.78 | 11.53 | |
| Poor rotamers (%) | | | 0.13 | 0.03 | 0.00 | 0.04 | |
| Ramachandran plot | | | | | | | |
| Favored (%) | | | 93.97 | 944.70 | 95.11 | 95.78 | |
| Allowed (%) | | | 6.00 | 5.24 | 4.71 | 4.03 | |
| Disallowed (%) | | | 0.04 | 0.06 | 0.18 | 0.19 | |

IFT-A structures is from subunit IFT122, whose structure defines how the two modules are connected in the two states (Fig. 3a, b). In the elongated state, the TPR array of IFT122 starts from the IFT121-IFT122 junction in the base module and then sticks into the core of the head module in a fully elongated manner (Fig. 3b). In stark contrast, the TPR array makes a sharp bending between the fourth and fifth repeats such that the entire head module folds back onto the base as a rigid body (Fig. 3b). The 11-residue loop between TPR repeats 4 and 5 of IFT122 (Loop-45, residues 707–717) that is disordered in both elongated and folded states functions as a hinge with high flexibility to accommodate the large conformational change of IFT-A during the transition between the two states (Fig. 3b and Supplementary Fig. 6b). Notably, in contrast to the periphery locations in the elongated state, the long TPR arms of IFT140 and IFT144 are encircled by the WDs of IFT140 and the TPRs of IFT121, IFT122 and IFT139 in the folded state (Fig. 3c). These intimate contacts consequently lead to the burying of ~1000 Å²

solvent-accessible surface area between the head and base modules, suggesting that the folded state is energetically more favorable than the elongated state (Fig. 3c).

To further investigate the dynamics of the head and base modules in the IFT-A complex, we calculated the range of the relative positions between the two modules by using the angular assignments of the particles from focused reconstructions. This analysis unveiled two major classes of IFT-A conformations with distinct features (Fig. 3d). The first class corresponds to the folded state with a narrow angular distribution, while the second one is the elongated state with a broader angular distribution in which the head and base modules could flex by ~30° relative to each other (Fig. 3d and Supplementary Movie 1). We propose that the elongated and folded conformations captured by our cryo-EM reconstruction represent two local energy minimum states of the apo IFT complex in the conformational space in aqueous solution. By contrast, only

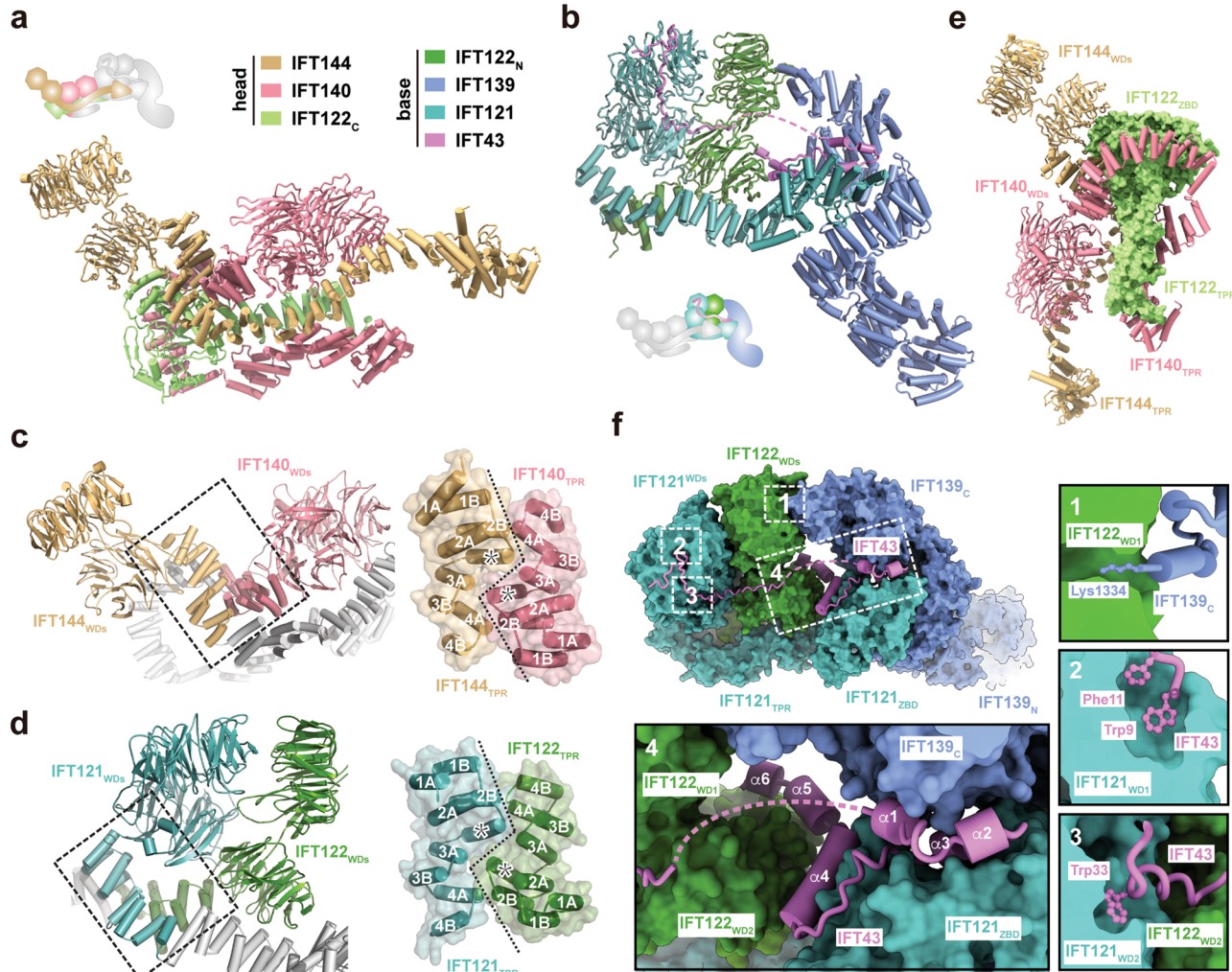

**Fig. 2 | The head and base modules of IFT-A.** Atomic structures of the head (**a**) and base (**b**) modules. The color scheme is shown. The cartoon indicates the relative position of the modules in the elongated state of IFT-A. The pseudo-two-fold symmetric IFT144-IFT140 (**c**) and IFT122-IFT121 heterodimers (**d**). The central TPR junctions formed by the first four TPRs are highlighted in black dashed boxes, which are enlarged on the right. The TPRs are numbered and the inserted α-helix in each subunit is labeled with an asterisk. **e** A different view of the head module with IFT122 shown in the surface representation. **f** Close-up views of the interaction network in the base module between IFT122$_{WD1}$ and IFT139$_C$ (1), between IFT121$_{WD1}$ and IFT43 (2), between IFT121$_{WD2}$ and IFT43 (3) and between IFT43 and IFT139-IFT121-IFT122 (4).

the elongated conformation with local fluctuations was identified in other IFT-A structures reported recently[27–30].

## Assembly of IFT-A into the anterograde IFT train

The atomic model of the apo IFT-A complex provides us a unique opportunity to understand the molecular basis of how the IFT-A complex is incorporated into the IFT train through polymerization. We set out to fit the IFT-A structures into the 20.7-Å cryo-ET density map of the anterograde IFT train from green alga *Chlamydomonas*[27]. Markedly, we found that the rigid head and base cores can be unambiguously fit into the anterograde train density individually with high cross-correlation scores of 0.80 and 0.78 respectively (Fig. 4a), suggesting that the apo IFT-A exists in a preformed, bimodular conformation before the incorporation into the train[11,13,14,38].

Calculation of the relative orientation between the head and base modules of IFT-A in the anterograde IFT train clearly unveiled that IFT-A in the anterograde train is more similar to the elongated than to the folded state (Fig. 3d). The elongated apo IFT-A can be docked into the cryo-ET density map of the anterograde train via a serial of conformational changes (Fig. 4a and Supplementary Movie 2). First, a slight tightening of the twist in IFT139$_N$ allows this

solenoid structure to snugly fit into the vacant volume in the bottom of the anterograde map and to contact IFT121$_{ZBD}$ from the neighboring IFT-A (Fig. 4a, b and Supplementary Fig. 10). This adjustment of IFT139$_N$ polymerizes the IFT-A complexes with an [-IFT139$_N$-IFT121$_{ZBD}$-]$_n$ repetitive pattern in the anterograde train (Fig. 4a, b). Meanwhile, the head core rotates ~50° counterclockwise around the Loop-45 hinge in IFT122 to fit into the corresponding position in the anterograde train, in which WD1 of IFT140 makes contacts with the neighboring WD1 of IFT121 at the −1 position (Fig. 4a, b). Then, the flexible TPR arms of both IFT140 and IFT144 untwist from their clustered positions near the central TPR arm of IFT122 to extend to opposite directions and associate with the TPR arms of IFT144 and IFT140 from the neighboring IFT-A complexes at the +1 and −1 positions, respectively (Fig. 4a, b). Iteration of this process establishes the tandem connection of IFT-A in the anterograde train in an arm-in-arm fashion (Fig. 4a, b). Strikingly, IFT144$_{ZBD}$ in the train also establishes a connection with IFT139$_N$ from the IFT-A complex at the −2 position, suggesting that the structural unit of IFT-A in the anterograde train consists of three consecutive interlinked IFT-A complexes (Fig. 4b). Consistent with this observation, two mutations in human IFT144, C1253Y and C1267Y (equivalent to C1286 and

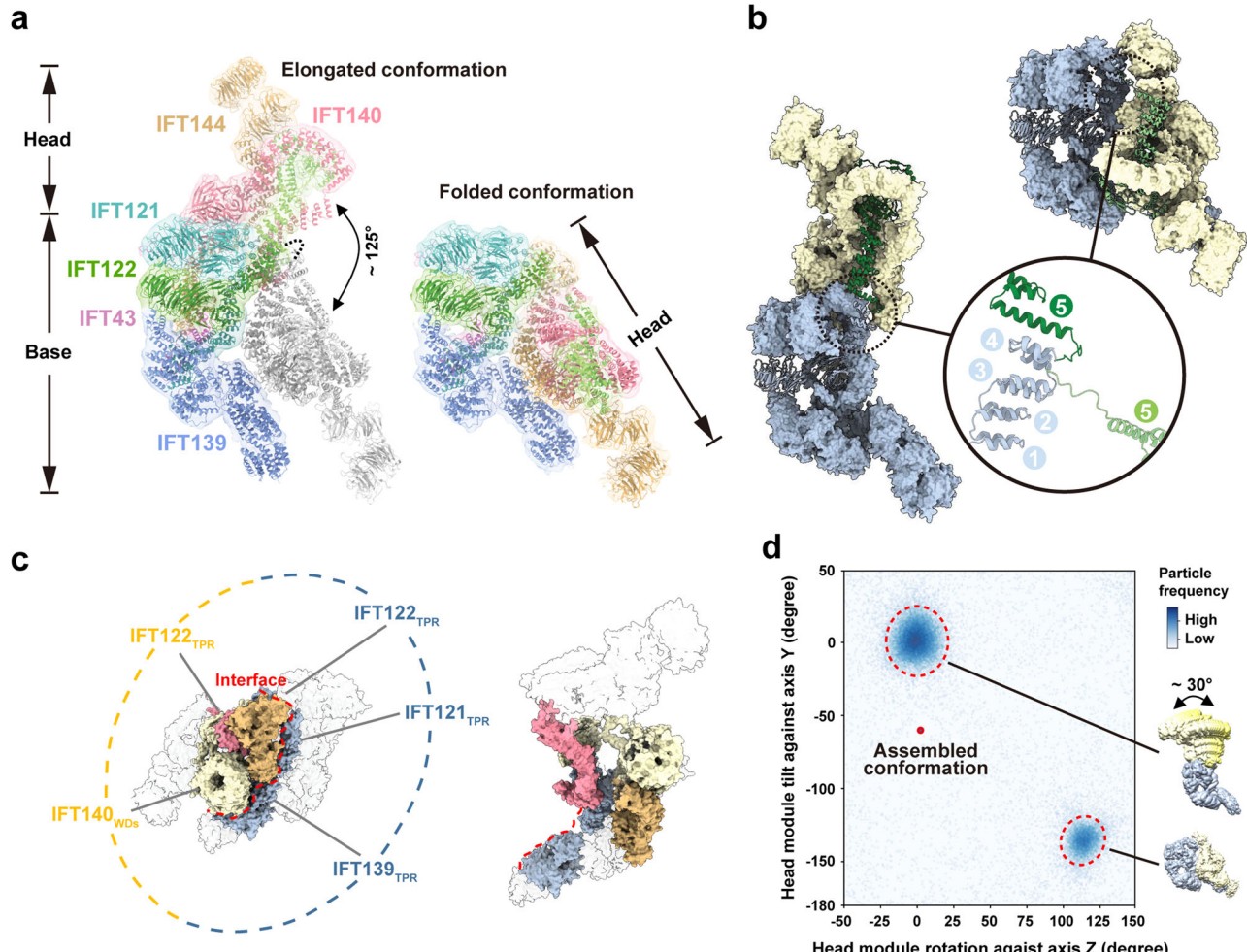

**Fig. 3 | Structural comparison of the elongated and folded states of IFT-A. a** The atomic models of the elongated (left) and folded (right) states of IFT-A. IFT-A proteins are colored same as Fig. 1a. The folded state is superimposed onto the elongated state based on the structure of the base module while its head module is colored in gray. **b** Comparison of the TPR arm of IFT122 in the two states. IFT122 is colored as Fig. 1a while the remaining portion of IFT-A colored as Fig. 1e. Super-position of the bending site of IFT122 in the two states. The first four TPRs are colored in pale blue while the fifth TPR and Loop-45 colored in green in the elongated state and in light green in the folded state, respectively. **c** The interface between the head and base modules in the folded (left) and elongated state (right). IFT140$_{TPR}$ and IFT144$_{TPR}$ are colored as Fig. 1a, IFT139$_{TPR}$, IFT121$_{TPR}$, IFT122$_{TPR}$ and IFT140$_{WD}$ as Fig. 1e, and the rest of IFT-A in light gray. **d** Plot of the head-base relative orientation showing two clusters that correspond to the elongated and folded states, respectively. The relative orientation between the two modules in assembled IFT-A in the anterograde train is denoted by a red cycle. The head movements in the elongated state are shown.

C1300 in *Tetrahymena* IFT144) that disrupt the zinc ion binding leads to Nephronophthisis and Jeune syndrome, underscoring the important role of IFT144$_{ZBD}$ in ciliary function[39,40]. Collectively, this polymerized, highly interconnected configuration in the anterograde train stabilizes IFT-A in a fully extended conformation that otherwise is unfavorable compared to the elongated and folded apo states of IFT-A (Fig. 4b).

This model unveiled two distinct IFT-A surfaces from the radical direction in the anterograde train. The exterior surface is enriched with WDs that abut the cilia membrane and play an essential role in the delivery of various membrane-binding proteins (Fig. 4c)[9,41-43]. The interior surface on the other side of IFT-A is characterized by convex-shaped TPR clusters from IFT140, IFT144 and IFT139 (Fig. 4c and Supplementary Fig. 11). To gain more insights into the interactions between IFT-A and IFT-B in the anterograde train, we built a pseudo-atomic model by docking our assembled IFT-A structure and the published IFT-B structure into a composite map of the anterograde IFT train (Fig. 4c)[27]. The model unveils three putative contacting areas between IFT-A and IFT-B, IFT140$^{IFT-A}$-IFT172$^{IFT-B}$, IFT144$^{IFT-A}$-IFT88$^{IFT-B}$, and IFT139$^{IFT-A}$-IFT74/IFT81$^{IFT-B}$ (Fig. 4c), which are consistent with previous biochemical studies[15,27,29,44-47]. This assembled configuration

of IFT-A in the anterograde train is in line with the notion that the structural unit of the anterograde trains is made of IFT-A, IFT-B and dynein-1b in a ratio of 3:6:2 (Fig. 4d)[15].

**Pathogenic mutations in IFT-A**

To date, there are more than 140 pathogenic missense mutations in IFT-A registered in the Human Gene Mutation Database (HGMD, http://www.hgmd.cf.ac.uk/) (Supplementary Figs. 6, 12 and Supplementary Data 2). The high sequence conservation of IFT-A proteins among different organisms allows us to investigate the pathogenic mechanisms of missense mutations in human IFT-A (Fig. 5). Structural analysis of the *Tetrahymena* IFT-A complex in the anterograde train reveals that these disease-causing mutations could be divided into four distinct groups: mutations in the WDs of IFT-A proteins (type I); mutations at the [IFT-A]-[IFT-A] assembly interface (type II); mutations at the [IFT-A]-[IFT-B] assembly interface (type III); and mutations at the component interface within the IFT-A complex (type IV) (Fig. 5 and Supplementary Fig. 12). Given that all the IFT-A WDs directly face the cilia membrane, the majority of type-I mutations in WDs likely has adverse effects on cargo trafficking directly or indirectly through

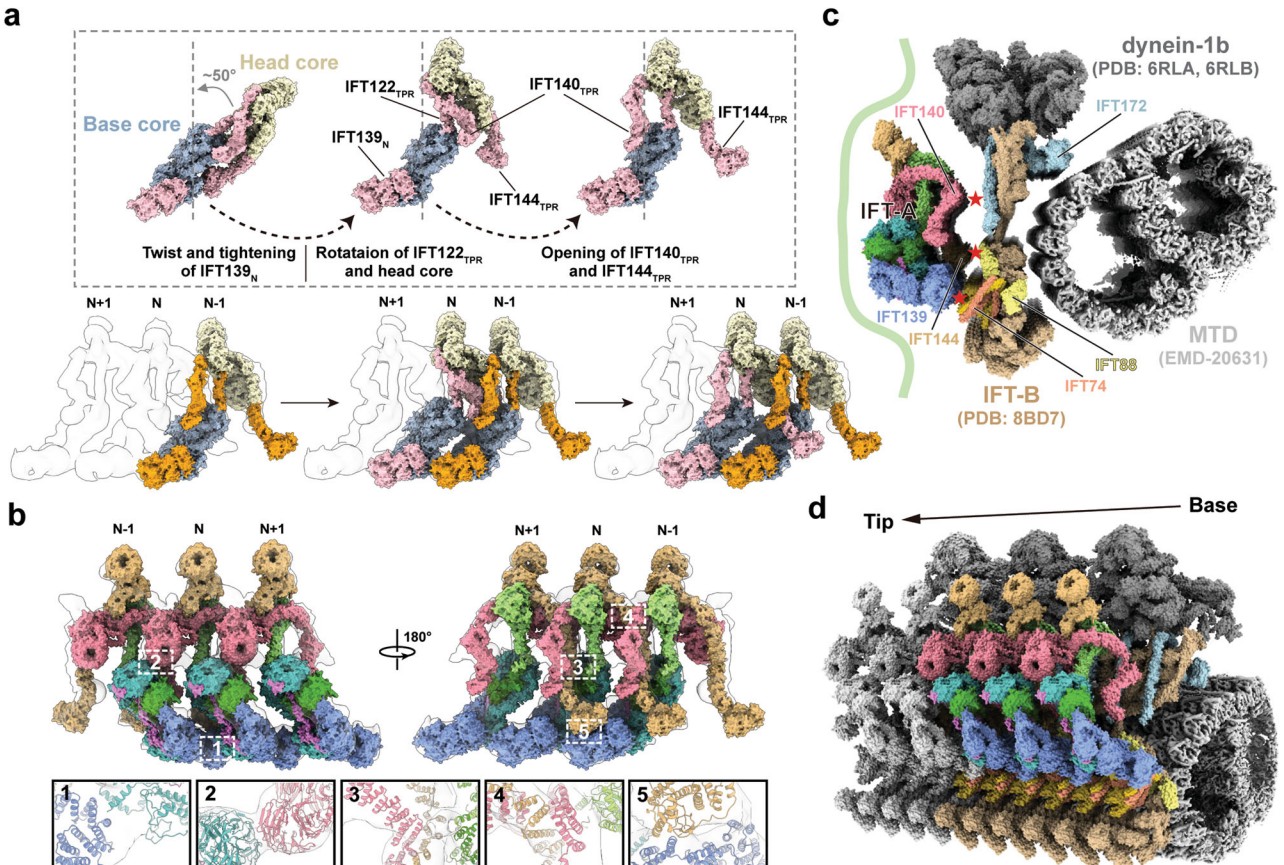

**Fig. 4 | Assembly of the IFT-A complex into the anterograde IFT train. a** Fitting of IFT-A into the cryo-ET density map of the anterograde train. The head and base cores are colored in light yellow and pale blue as Fig. 1e, and the flexible TPRs in the elongated and assembled states colored in pink and orange, respectively. **b** Close views of the inter-molecule interactions formed between IFT121$_{ZBD}$ and IFT139$_N$ (1), between IFT140$_{WD}$ and IFT121$_{WD}$ (2), between IFT144$_{TPR}$ and IFT140$_{TPR}$ (3-4), and between IFT144$_{ZBD}$ and IFT139$_N$ (5). **c** Fitting of atomic models of *Tetrahymena* IFT-A and *Chlamydomonas* IFT-B into the anterograde train map unveiling the putative

[IFT-A]-[IFT-B] interfaces. IFT-A is from this study and colored as Fig. 1a, and IFT-B is from Lacey et al. with IFT172 (residues 1-1104) colored in cyan, IFT88 (residues 122-690) in yellow, IFT74 (residues 135-340) in orange and the rest of IFT-B in light brown. EMDB code of microtubule doublet (MTD) is EMD-20631. PDB codes of IFT-B, dynein-2 motor domain and tail domain are 8BD7, 6RLA and 6RLB, respectively. **d** Periodic arrangement of the anterograde train. Six IFT-B, three IFT-A and two dynein-1b in one periodic unit are colored as **c**, while the remaining is shown in grey.

impairing the folding and/or stability of WDs (Figs. 4c and 5). By contrast, mutations at type-II and -III interfaces would interfere with IFT-A polymerization into the anterograde IFT train (Figs. 4c, 5 and Supplementary Fig. 12), while mutations of type IV would lead to the instability of the IFT-A complex (Figs. 2a, b and 5). Notably, all type-III mutants are located either in IFT144$_C$ or in IFT139$_N$, in line with the observation that both IFT144 and IFT139 are at the interface between IFT-A and IFT-B (Supplementary Fig. 12). Intriguingly, human IFT139, in spite of its periphery position in IFT-A and lacking WDs, is a frequently disease-causing subunits among all IFT-A proteins (Fig. 5, Supplementary Figs. 6, 12 and Supplementary Data 2). In particular, mutations that cause the Meckel-Gruber syndrome (MKS), a lethal autosomal recessive congenital anomaly disease, are only found in IFT139 (Fig. 5 and Supplementary Data 2), underscoring the essential role of IFT139 in cilia function[21,38,48].

## Discussion

In this study, we determined the cryo-EM structure of the IFT-A complex that provides us a unique opportunity to understand the architecture of IFT-A, its assembly in the IFT trains and its role in motor-driven cargo transport within cilia. IFT-A proteins are found both in the anterograde and retrograde trains as well as in the cytoplasm with the most concentrated distribution at the cilia base[5,18,49]. It has been unclear whether these IFT-A components assemble the IFT-A complex on the time when

they are incorporated into the IFT trains or they preassemble the IFT-A complex in the cytoplasm first before the incorporation into the trains. The purification and visualization of endogenous *Tetrahymena* IFT-A complexes reported here suggest that the IFT-A complex very likely exists as a preassembled bimodular complex at the cilia base (Fig. 1). Consistently, in a recent study recombinantly expressed human IFT-A proteins were mixed to obtain the human IFT-A complex whose structure, although only partially resolved at the atomic level, is similar to the elongated conformation of the *Tetrahymena* IFT-A complex, supporting the notion that IFT-A components could spontaneously form the IFT-A complex in solution before incorporation into the IFT trains (Fig. 1)[29]. This highly efficient, hierarchical mechanism is a common feature for biomacromolecule assembly, a good example of which is the nuclear pore complex, whose formation involves a hierarchy of assembly steps of well-defined subcomplexes[50].

During the stepwise assembly of the anterograde IFT train at the ciliary base, the self-polymerized IFT-B array first attaches to the microtubule, and then the IFT-A complex is incorporated into the train on the backbone of IFT-B facing the cilia membrane (Fig. 6)[31]. It was reported that IFT74[IFT-B] and IFT139[IFT-A] are required for the association between IFT-A and IFT-B[15,47]. Concordantly, model fitting of our high-resolution IFT-A structure into the in situ density map of the anterograde train from green alga *Chlamydomonas* unveils that IFT74[IFT-B] is in close vicinity to IFT139[IFT-A] (Fig. 4c and Supplementary Fig. 11)[15,27]. We propose that the preassembled IFT-A initiates the mounting onto

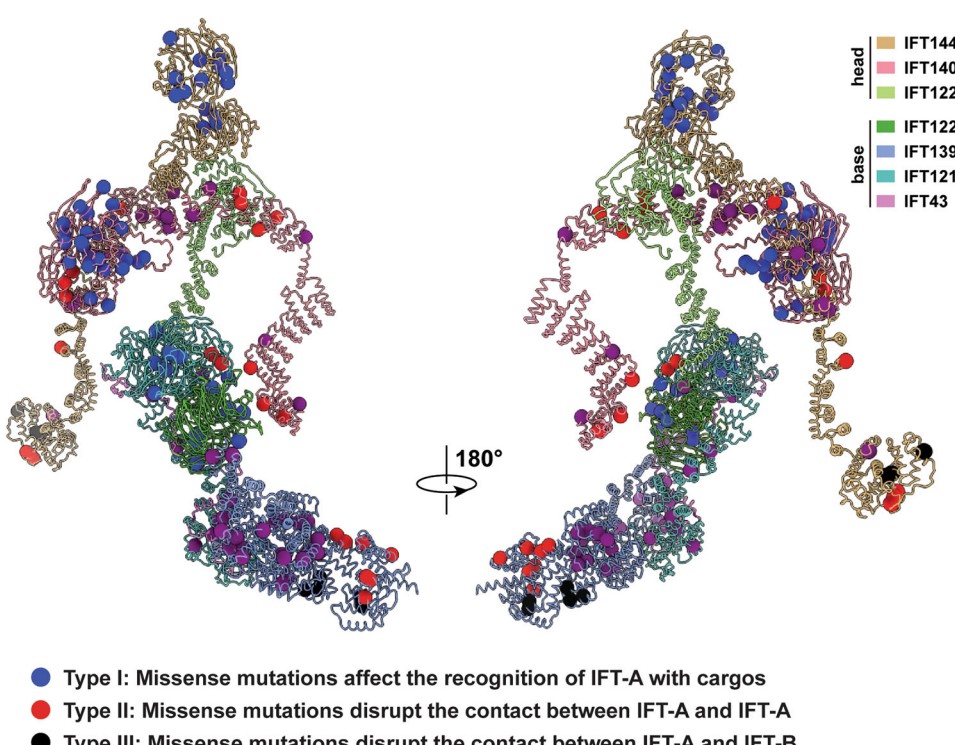

| | |
|---|---|
| head | IFT144 |
| | IFT140 |
| | IFT122$_C$ |
| base | IFT122$_N$ |
| | IFT139 |
| | IFT121 |
| | IFT43 |

● **Type I:** Missense mutations affect the recognition of IFT-A with cargos
● **Type II:** Missense mutations disrupt the contact between IFT-A and IFT-A
● **Type III:** Missense mutations disrupt the contact between IFT-A and IFT-B
● **Type IV:** Missense mutations impair the interaction of the components within IFT-A

**Fig. 5 | Summary of missense mutations in the IFT-A complex.** The pathogenic mutations of IFT-A can be divided into four distinct categories (type I, II, III and IV) based on their roles in cargo delivery or IFT train assembly.

the IFT-B scaffold through the interaction between IFT74[IFT-B] and IFT139[IFT-A], which induces the twist of the N-terminal TPRs of IFT139[IFT-A] to allow IFT139[IFT-A] to interact with adjacent IFT121[IFT-A] to polymerize all the IFT-A base modules in the train (Fig. 4a and Supplementary Fig. 11)[27]. In addition to the IFT139[IFT-A]-IFT74[IFT-B] contact in the IFT-A base module, IFT-A fitting into the anterograde train also reveals spatial proximities between IFT88[IFT-B] and IFT144[IFT-A] and between IFT172[IFT-B] and IFT140[IFT-A] (Fig. 4c and Supplementary Fig. 11)[29,44–46]. We speculate that these two contacts between IFT-A and IFT-B accompany a cascade of events to complete the incorporation of IFT-A into the anterograde train, including the rotation of IFT122 stem, the untwist of TPR arms of IFT140 and IFT144 as well as the establishment of the IFT140-IFT144 connection between adjacent IFT-A complexes in an arm-in-arm fashion (Fig. 4b). Higher-resolution in situ structural information of the anterograde IFT train is required to further verify this model.

Once reaching the ciliary tip, the densely packed anterograde IFT trains are remodeled into the retrograde trains that adopt a loose zigzag shape in raw tomograms[15–19,31,51]. How are the same IFT-A and IFT-B complexes transformed between these two states with markedly distinct conformations? A salient, common structural feature for both IFT-A and IFT-B is their bimodular architectures with a flexible hinge region in the middle (Fig. 1 and Supplementary Fig. 11)[27]. Similar to the head and base modules of IFT-A, IFT-B consists of two stable sub-complexes IFT-B1 and IFT-B2 as revealed by the 9.9-Å and 11.5-Å in situ cryo-ET density maps, respectively (Supplementary Fig. 11)[27]. Given the stability of these modules, it is unlikely that IFT-A and IFT-B in the anterograde train are completely disassembled into individual sub-units before the reassembly of the retrograde train (Fig. 6). Instead, we propose that, with the aid of the active configuration of dynein-1b, IFT-A and IFT-B more likely make bending at the hinge regions combined with twists of the flexible TPRs in the IFT proteins to transform the anterograde train into the zigzagged retrograde train while maintaining the internal structures of each module in IFT-A and IFT-B (Fig. 6).

In summary, high-resolution atomic models of IFT-A presented here and by others reveal highly similar elongated conformations, providing valuable insights into the assembly of the anterograde IFT train[27–30]. A unique finding of our study is the identification of the folded state of IFT-A that is clearly distinct from those of both elongated and anterograde-train assembled IFT-A conformations. We hypothesize that this folded conformation is a state related to IFT-A in the retrograde train or it represents a state at the ciliary tip before the assembly of retrograde IFT trains (Fig. 6). Future high-resolution in situ structural studies are needed to answer these questions and to understand the transportation mechanisms of both anterograde and retrograde IFT trains.

## Methods

### Cell line establishment

*Tetrahymena thermophila* strains B2086, CU428.2 and plasmid pFZZ-neo4 were purchased from Tetrahymena Stock Center (Cornell University). FZZ tag in the plasmid was separated by a tobacco etch virus protease (TEV) cleavage site between Flag peptide (F) and tandem Protein A domain modules (ZZ). The last ~1000 bp of *IFT122* coding sequence and ~1,000 bp of *IFT122* 3′ untranslated region (UTR) were separately amplified from *Tetrahymena* genomic DNAs and cloned into the pFZZ-neo4 vector. The primers are listed as follows:

Fw (IFT122) 5′-CGCTCTAGAACAACCTAATAAAAAGAAGAAGAAAGCG-3′,

Rev (IFT122) 5′-ATCGGATCCGAAATCAAAAACATCCTTCTAAGGTCC-3′,

Fw (IFT122 3′UTR) 5′-TCAAGCTTAGCAAAAGCCTACAAATAATTAAAAG-3′,

Rev (IFT122 3′UTR) 5′-CCCCTCGAGACACAAGACTTCGCACATAAAATG-3′.

Then, the DNA fragment encompassing IFT122-FZZ-neo4 was digested by double endonucleases from the vector, injected into conjugants of B2086 and CU428.2 by electro-transformation and

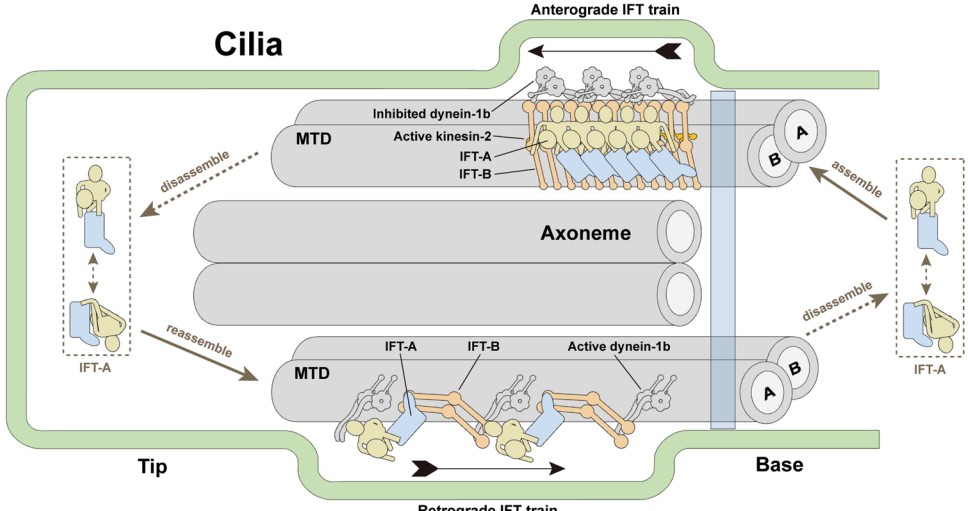

**Fig. 6 | Schematic diagram of IFT-A cycle.** IFT-A complexes from the assembly of freshly expressed IFT-A subunits and the disassembly of retrograde IFT trains are concentrated in the basal bodies. Preassembled bimodular IFT-A are incorporated into anterograde IFT train by bending at the hinge regions and twists of the flexible TPRs. After reaching the tip, IFT-A and IFT-B dissociate from anterograde train, likely through adjustments at the hinge regions and the flexible TPRs to transform the anterograde train into the loosely zigzagged retrograde train. At last, retrograde trains move back to the basal bodies and IFT-A detaches from the retrograde train to finish a cycle. Opening questions about the mechanism of ciliary tip turnaround and retrograde train assembly, as well as the roles of folded IFT-A complex in IFT-A cycle await further studies.

integrated at the endogenous *IFT122* locus. Then the exconjugants were transferred to fresh Neff medium and transformants were selected by gradually increasing concentration of paromomycin. When cells were unable to double within 24 hours, single cell was sorted out by BD Influx and cultured in fresh medium without paromomycin for one week. Then we extracted genomic DNAs, verified genotypes of the cells by quantitative real-time PCR (RT-qPCR) and chose clones with total gene replacement for large-scale culture.

### Purification of the IFT-A complex

For affinity purification, 20 L of cells expressing FZZ-IFT122 were cultured in Neff medium at 30 °C to mid-log phase (4 × 10$^5$ cells per mL). Then cells were harvested and lysed in a buffer containing 50 mM HEPES (pH 7.4), 50 mM NaCl, 10% (v/v) glycerol, 0.05% (v/v) IGEPAL CA-630 and 1 × protease inhibitor cocktail (MedChem Express). After sonication and spinning, supernatant was collected and incubated with 400 μL IgG Sepharose 6 Fast Flow beads (GE Healthcare) at 4 °C for 6 hours. Then, the beads were washed with lysis buffer and digested overnight by TEV protease. The eluate from the IgG beads were bound to anti-DYKDDDDK G1 affinity resin (GenScript) for 2 hours. After washing with lysis buffer, the IFT-A complex protein was eluted with 2 mL of 0.2 mg mL$^{-1}$ Flag peptide in a buffer containing 50 mM HEPES (pH 7.4), 50 mM NaCl and 0.025% (v/v) IGEPAL CA-630 and concentrated to 20 μL at 3 mg mL$^{-1}$. Silver staining SDS-PAGE gel, native-PAGE gel and mass spectrometry were performed to analyze the purity and homogeneity of the IFT-A complex.

### Cryo-EM grid preparation and data collection

Cryo-EM grids were prepared with the Vitrobot Mark IV plunger (FEI) set to 8 °C and 100% humidity. 2.5 μL of protein sample was applied onto a glow-discharged Au Quantifoil R1.2/1.3 300 mesh holey carbon grid. Immediately afterwards, a blot force of −1 and blot time of 3 seconds were applied to blot the grids. Then the samples on grids were vitrified by plunge freezing in pre-cooled liquid ethane at a liquid nitrogen temperature.

Frozen-hydrated grids were loaded into a 300 kV FEI Titan Krios G3i electron microscope equipped with Gatan K3 direct electron detector, Data were automatically acquired with EPU software (FEI Eindhoven, the Netherlands). Gain-normalized movies were collected at a magnification of 81,000 in super-resolution mode (pixel size of 0.55 Å) and at a defocus range between −1.5 and −2.5 μm. The total dose of 50 e$^-$/Å$^2$ was fractionated into 32 frames. In order to improve the resolution, a total number of 66,729 movie stacks were acquired for the structures presented in this work.

### Cryo-EM image processing

Global and local motion correction as well as dose weighting were performed for each frame of original super-resolution movie stacks by MotionCor2[52]. Both dose-weighted and unweighted micrographs were resized to a pixel size of 1.1 Å with a binning factor of two. The dose-weighted micrographs were used for particle picking and further data processing, and the non-dose-weighted ones were subjected to Contrast Transfer Function (CTF) parameters searching in Gctf[53]. The outliers of pre-processing results were manually filtered out by checking the motion and CTF fitting qualities.

For the first dataset, an initial group of ~1000 particles are manually picked and applied to reference-free 2D classification in RELION 3.0[54]. The resultant 2D class averages were used as templates for automatic particle picking in Gautomatch (https://www2.mrc-lmb.cam.ac.uk/download/gautomatch-056/). All particles were normalized and binned by four during extraction with a box size of 128 pixels. Several rounds of 2D classification jobs were performed to remove contaminants and noisy particles. Then the output particles were subjected to a large amount of 3D classification jobs with different combinations of class number and regularization parameter T. All the particles corresponding to the density maps of higher resolutions and lower noise levels were merged with duplicates removed, and then re-extracted with a pixel size of 1.1 Å. The following high-resolution processing steps including masked refinement, 3D classification without alignment as well as local angular searching yield to a density map which proved to be the base module of IFT-A at a resolution of 4.8 Å based on the gold-standard Fourier Shell Correlation (FSC) cut-off criterion of 0.143. A new round of particle picking was iterated for all datasets by using projections of this newly obtained density map. The new particle set were binned to a pixel size of 4.4 Å, cleaned by rounds of 2D classification and finally subjected to 3D classification with a larger particle diameter of 320 Å. As a result, all density maps including the base module of IFT-A and full IFT-A complexes in elongated and

folded states were obtained. With optimization of the mask diameter of particles, particle recentering and further 3D classification, the overall resolutions of the maps for the base module, IFT-A in folded state and IFT-A in elongated state were finally improved to 3.6 Å, 8.8 Å and 8.5 Å, respectively. After Bayesian polishing and CTF refinement, the density maps of the head module were improved to 4.2 Å (IFT140$_{1-1081}$ traceable) and 4.6 Å (full-length IFT140 traceable), respectively. For better visualization of the flexible IFT139$_N$ (residues 1–694) in the base module and IFT144$_C$ (residues 1130–1387) in the head module, the base and head density maps were finally resolved at 4.7 Å and 6.0 Å, respectively, by signal subtraction, classification without alignment and local angular searching. Local resolution variations are estimated by ResMap[55].

### Model building

We combined de-novo model building and rigid-body docking to generate the atomic models of the IFT-A complex. In order to manually build models of IFT-A in Coot, we focused on the unique residues with large sidechains at the beginning and end of each domain for component assignment. Aromatic residues at the beginning (residues 1–10) and the end (residues 331–340) of IFT121$_{WD1}$ allowed us to distinguish IFT121 from IFT122. Similarly, the last β-sheet of IFT140$_{WD2}$ (residues 705–714) allowed us to distinguish IFT140 from IFT144. The continuity of density maps of the base module at 3.6 Å and the head module at 4.2 Å allowed us to unambiguously trace most of the IFT-A residues. Due to the flexibility, AlphaFold-2 was employed to assist model building for IFT139$_N$ (residues 1–694) of the base module, IFT144$_{WD1}$, IFT144$_C$ (residues 1130–1387) and IFT140$_C$ (residues 1082–1407) of the head module[32]. The final model of the entire complex was iteratively refined by PHENIX and validated by MolProbity[56,57]. Data collection and refinement statistics were listed in Table 1.

We used the 20.7-Å in situ cryo-ET map of IFT-A deposited in Electron Microscopy Data Bank (EMDB) and our structure of IFT-A in the elongated state to build a pseudo-atomic model of the assembled IFT-A polymer[27]. Due to the stability of the head and base cores and the flexibility of the TPRs, we firstly truncated the flexible TPRs including IFT122 (residues 707–837), IFT139 (residues 1–766), IFT140 (residues 1035–1407) and IFT144 (residues 966–1387) from the structure of IFT-A in the elongated state. The unique conformations of the head and base module cores allowed us to place them into the cryo-ET map of IFT-A unambiguously with high cross-correlation scores (0.80 and 0.77, respectively). Then, we manually adjusted the flexible TPRs without changing their topologies to fit into the density and built the structural model of the assembled IFT-A. Lastly, multiple copies of the assembled IFT-A model were docked into the IFT-A map. To explore the interaction network between IFT-A and IFT-B, a composite map of the anterograde IFT train was generated by docking IFT-A (at 20.7 Å, EMD-15980) and IFT-B (at 9.9 Å, EMD-15977) into an assembling IFT train (at 29.9 Å, EMD-15261)[27,31]. Microtubule (EMD-20631) was placed referring to a previous study[15]. Then, structural models of the assembled IFT-A (this study), IFT-B (PDB-8BD7) and human dynein-2 (PDB-6RLA and 6RLB, dynein-1b in *Chlamydomonas reinhardtii*) were subsequently fit into the composite map to generate a pseudo-atomic model of the anterograde IFT train. Figures of the EM density maps and atomic models were generated using UCSF ChimeraX and PyMol[58].

### Quantification of head-base relative orientations

For the elongated and folded states, two clusters of particle images to obtain the corresponding density maps were used to calculate the distribution of relative orientations between the head and base modules. Each cluster of particles were recentered and re-extracted on both the head and base according to the results of previous consensus refinements. By taking the map of the elongated state which was low-pass filtered to 10 Å as reference, masked refinements on the recentered particle images were performed to respectively reconstruct the head and base maps. As a result, each particle was reassigned with two groups of Euler angles and then, the corresponding rotation matrices for the head and base maps were obtained. The rotation matrix indicating the relative rotation from head to base was calculated by the production of the rotation matrix for the head and the inverse rotation matrix for the base. Based on this matrix, Euler angles were obtained in ZYX sequence. A previous study also employed a similar quantification method[59]. For the state of the assembled IFT-A in the anterograde train, the rotation matrices for the head and base were generated by using ChimeraX to fit the corresponding parts of the anterograde train model to the density map of the elongated state[58].

### Reporting summary

Further information on research design is available in the Nature Portfolio Reporting Summary linked to this article.

## Data availability

The cryo-EM maps of IFT-A complex have been submitted to the Electron Microscopy Data Bank under accession numbers: EMD-34893 (IFT-A in elongated state at 8.5 Å), EMD-34894 (IFT-A in folded state at 8.8 Å), EMD-34895 (base module of IFT-A at 3.6 Å), EMD-34896 (base module of IFT-A at 4.7 Å), EMD-34897 (head module of IFT-A at 4.2 Å), EMD-34898 (head module of IFT-A at 4.6 Å) and EMD-34899 (head module of IFT-A at 6.0 Å), and the atomic coordinates have been deposited to the Protein Data Bank with accession codes 8HMC (base module of IFT-A at 3.6 Å), 8HMD (base module of IFT-A at 4.7 Å), 8HME (head module of IFT-A at 4.2 Å) and 8HMF (head module of IFT-A at 4.6 Å). Additional details on datasets and protocols that support the findings of this study will be made available by the corresponding author upon reasonable request. Source data are provided with this paper.

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

## Acknowledgements

We thank the staff of the Electron Microscopy Facility and Mass Spectrometry Facility at Shanghai Institute of Precision Medicine for providing technical support and assistance during data collection. We thank Wenyan Huang at Shanghai Children's Hospital, Shanghai Jiaotong University School of Medicine for communications about IFT-A diseases. This work was supported by the National Key R&D Program of China (2016YFA0501800 to Y.M. and 2018YFA0107004 and 2018YFC2000102 to M.L.), the National Natural Science Foundation of China (31930063 to M.L., 32071189 to J.W. and 31971137 to C.H.), the Innovative Research Team of High-level Local University in Shanghai (SHSMU-ZLCX20211700 to M.L., J.W. and Y.M.) and the Shanghai Municipal Education Commission Gaofeng Clinical Medicine Grant Support (20181711 to J.W.).

## Author contributions

M.L. and J.W. conceived the project. Y.M purified the IFT-A complex, prepared cryo-EM specimens and collected datasets. Y.M. and S.L. determined the structures. J.H., D.Y. and J.W. carried out model building and refinement. All authors contributed to data interpretation. J.W. made a major contribution in identifying the folded state of IFT-A. And M.L. Y.M. C.H. and J.H. wrote the manuscript.

## Competing interests

The authors declare no competing interests.
