## [Peer Review File · Nature Communications]

Structural insight into the intraflagellar transport complex IFT-A and its assembly in the anterograde IFT trainREVIEWERS' COMMENTS

Reviewer #1 (Remarks to the Author):

IFT (Intraflagellar transport) transports ciliary/flagellar component proteins from the cytoplasm to the tip. It consists of two protein complexes, IFT-A and IFT-B, which form periodic arrays. When IFT heads toward the tip (anterograde IFT), IFT-A is responsible to carry IFT dynein, which will be used for retrograde IFT from the tip to the cytoplasm. Ma and colleagues built atomic model of the IFT-A complex, employing single particle cryo-EM technique. The structure of IFT-A solved by them using specimen isolated from *Tetrahymena* cilia shows coexistence of two conformations (elongated and folded), both of which are made of two parts, which they call head and base modules. Most component proteins have repeating TPR motifs. The difference between these two conformations is located at IFT122, refolding as a pivot of the conformational change of the entire complex. They found that the elongated conformation fits to the subtomogram average from the past cryo-ET work on the intact IFT.

It should be noted that two preceding works on IFT-A structure were published recently. One (Hesketh et al. 2022 Cell 185, 4971, published on Dec/22/2022) used IFT-A reconstituted from individual component proteins, while the other Lacey et al. (2022) Nat. Struct. Mol. Biol. <https://doi.org/10.1038/s41594-022-00905-5>, published on Jan/2/2023 reconstructed IFT-A structure by cryo-ET of intact cilia and subtomogram averaging. The authors mentioned these two papers as in BioRxiv (Refs. 41&45), but they should be referred as the final published form. Interestingly these preceding works did not mention two coexisting structures. Although Ma et al. was preceded, since the authors used different specimen preparation method than the other two and showed unique “folded” structure, this manuscript deserves publication in Nature Communications. They also achieved higher spatial resolution than others, especially at the head part (called IFT-A1 in Hesketh et al.).

While the preceding works should not prevent publication of this manuscript (it was submitted before publication of the others in the journals), careful comparison and interpretation of them will help to make this manuscript more attractive. For example, their two conformations are more apart than the fluctuations described in Hesketh et al. (see their Fig.1D). Please confirm this and if so, do the authors have any thought why Hesketh et al. did not see the second conformation?

Another novel finding, due to improvement of resolution, is conformation of IFT43 in the head module of IFT-A. This reviewer would recommend to show the detailed density map of IFT43 on other components to prove the loop is visible in the single particle cryo-EM analysis. In Hesketh et al. IFT43 was visible only at a short fragment and they employed AlphaFold-multimer to model the rest of IFT43 (Fig.2B). But this modeling is different from experimentally solved structure in this work. Was conformation predicted by AlphaFold2 wrong?

Other minor points:

Fig.1A: It should be shown which part of the sequence was solved in this structural study.

Show validation map (PAE plot) of Alphafold2.

Line118, 121, 131: it is not clear how Fig.5 is cited as a reference for pseudo-symmetry of IFT140/144 and 120/121.

Extended data figure 3: caption is too simple. Each panel must be identified.

Reviewer #2 (Remarks to the Author):

In this paper, the authors obtained the IFT-A structure from *Tetrahymena*. They did affinity pulldown purification and then cryo-EM to obtain the structures at different conformation from 3.6 - 9 Angstrom resolution. Using AlphaFold and modelling, they presented the model of the complete IFT-A complex and analysed the important interaction in the complex in regard to known mutations affecting IFT structure and function. They also compared their structure with the known in situ structure of *Chlamydomonas* IFT to show the conformation change in the IFT-A.

As someone in the field, I think this is a tremendous work from the authors to obtain the structure, which is elusive for a long time. The paper is presented very nicely in terms of figures and writing. My major concern is that the authors need to show how they modelled better because the resolution of the map is not easy to modelling. In addition, with the limitation of the resolution, the author must justify that the interpretation of the interaction at the amino acid side chain level throughout the paper is appropriate.

NOTE: There are 4 papers presenting the structure of IFT-A recently. I don't think they diminish the work of the authors but it is reasonable to mention or compare them with the structure from the authors.

My comments are as follows:

While the authors claim 3.6, 4.2 and 4.6 in some map, the map doesn't show the feature very well, especially the 4.6 Angstrom map, the helix are not very well separated, perhaps due to the threshold being too low for rendering. Is this possible to show the model well fitted in the map to validate the

feature there and probably the side chain fitting visualization in the 3.6 Angstrom map. Somehow in Figure S4 but a bit better highlighting.

With that resolution, how did the author distinguish IFT144 and IFT140 or IFT121 from IFT122 for modelling?

Also, with the modelling at 4.2 and 4.6 resolution, how does the author justify their interpretation of side chain interaction like in Figure 2f or Fig. S5c

Fig. 4a Opening of IFT140TPR is not clear from 3 transition figures.

It is important to label directly in Figure which one is not from this paper such as in Extended Figure 10.

Reviewer #3 (Remarks to the Author):

The manuscript by Ma et al provides important structural insights into the intraflagellar transport complex A (IFT-A) and its assembly in the anterograde IFT trains. This is a well written and thorough study describing novel conformations of IFT-A and provides detailed insights into how IFT-A monomers could incorporate into trains. The likely biological significance of the folded conformation in retrograde IFT, although speculative, is another highlight of the paper. Considering other papers in the field being published, this paper should be published immediately without any delay.

My comments below are mostly editorial and should be considered by the authors as they feel necessary.

1. Fig 2: IFT144 and IFT140 colors are too similar and could be changed.

2. Fig. 4. IFT-A and B interactions, including that between IFT74 and Thm1(IFT139) can be better highlighted in text in the results and figures. The authors might consider including details from Fig S10 in Fig 4c.

3. Fig. 4. IFT-A and B interactions, where do the authors think that the IFT140 interaction is happening with IFT-B? Based on Fig S10, the authors marked IFT172 close to IFT140, and if so, the authors should probably state it in the text.

4. Fig. 4. IFT-A and B interactions, where do the authors think that the IFT144 interaction is happening with IFT-B? Please comment in text if the authors think that IFT88 is involved or not.

5. Extended Fig 9. This figure is very speculative, but again is a nice addition to the paper and could be included in the main figures. The IFT139 regions interacting with IFT74 could be highlighted in text and figures, in line with the description of the type III mutations. Same for type III mutations described in IFT144 and IFT140, if any.

6. Lines 304-5: The results showing IFA-B subunit interactions should be introduced in this part of the Results section for clarity.

7. Lines 215-217: The sentence can be simplified.

8. Line 164: the authors might consider showing the seven IFT43 helices. I don't see them currently shown clearly in Extended Fig. 4.

Point-to-point responses to reviewers' comments

Reviewer #1:

IFT (Intraflagellar transport) transports ciliary/flagellar component proteins from the cytoplasm to the tip. It consists of two protein complexes, IFT-A and IFT-B, which form periodic arrays. When IFT heads toward the tip (anterograde IFT), IFT-A is responsible to carry IFT dynein, which will be used for retrograde IFT from the tip to the cytoplasm. Ma and colleagues built atomic model of the IFT-A complex, employing single particle cryo-EM technique. The structure of IFT-A solved by them using specimen isolated from Tetrahymena cilia shows coexistence of two conformations (elongated and folded), both of which are made of two parts, which they call head and base modules. Most component proteins have repeating TPR motifs. The difference between these two conformations is located at IFT122, refolding as a pivot of the conformational change of the entire complex. They found that the elongated conformation fits to the subtomogram average from the past cryo-ET work on the intact IFT.

1. It should be noted that two preceding works on IFT-A structure were published recently. One (Hesketh et al. 2022 Cell 185, 4971, published on Dec/2/2022) used IFT-A reconstituted from individual component proteins, while the other Lacey et al. (2022) Nat. Struct. Mol. Biol. <https://doi.org/10.1038/s41594-022-00905-5>, published on Jan/2/2023 reconstructed IFT-A structure by cryo-ET of intact cilia and subtomogram averaging. The authors mentioned these two papers as in BioRxiv (Refs. 41&45), but they should be referred as the final published form.

Corrected.

2. Interestingly these preceding works did not mention two coexisting structures. Although Ma et al. was preceded, since the authors used different specimen preparation method than the other two and showed unique “folded” structure, this manuscript deserves publication in Nature Communications. They also achieved higher spatial resolution than others, especially at the head part (called IFT-A1 in Hesketh et al.).

While the preceding works should not prevent publication of this manuscript (it was submitted before publication of the others in the journals), careful comparison and interpretation of them will help to make this manuscript more attractive. For example, their two conformations are more apart than the fluctuations described in Hesketh et al. (see their Fig.1D). Please confirm this and if so, do the authors have any thought why Hesketh et al. did not see the second conformation?

Thanks for this good point. Compared with recently reported single-particle cryo-EM IFT-A structures (at a resolution range of 3.4 to 8 Å)¹⁻⁴, the novel finding of our study is the identification of two different states of IFT-A, elongated and folded states, whereas other studies only present the IFT structure corresponding to the elongated conformation.

In all the elongated IFT structures reported recently, local fluctuations were observed both inside (structural variations in the flexible TPRs of IFT122, IFT139_N, IFT140 and IFT144) and between modules (rotation between the base and head modules). Similar fluctuations were also observed in the elongated IFT structure in this study. In stark contrast, the conformation changes between the elongated and folded states reported here are strikingly distinct from these local fluctuations. In the folded state, the head module folds back onto the base module to form a very compact structure. We hypothesize that the observed folded IFT-A structure likely is related to IFT-A in the retrograde train or represents a state at the ciliary tip before the assembly of the retrograde train.

The state-of-art unsupervised classification algorithm is still not optimized for the non-dominant group, especially when the corresponding initial reference is unknown. Hence, we believe that extensive classifications and careful removal of only obvious junk class were essential for us to identify the folded state. Meanwhile, image binning was also helpful for us to distinguish the elongated and folded states because of higher SNR in low spatial frequency and larger range of translational parameter searching.

Following this reviewer' suggestion, we have emphasized the novel finding of the folded IFT state of our study in the revised manuscript.

“Analysis of these structures reveal that the apo IFT-A complex is composed of two rigid structural modules, the head and base modules, and presents two distinct—elongated and folded states in solutions, the latter of which was not identified in recent IFT-A structural studies²⁷⁻³¹.” (Page 4, lines 64-67)

“We propose that the elongated and folded conformations captured by our cryo-EM reconstruction represent two local energy minimum states of the apo IFT complex in the conformational space in aqueous solution. By contrast, only the elongated conformation with local fluctuations was identified in other IFT-A structures reported recently²⁷⁻³⁰.” (Page 11, lines 203-207)

“In summary, high-resolution atomic models of IFT-A presented here and by others reveal highly similar elongated conformations, providing valuable insights into the assembly of anterograde IFT trains²⁷⁻³⁰. A unique finding of our study is the identification of the folded state of IFT-A that is clearly distinct from those of both elongated and anterograde-train assembled IFT-A conformations. We hypothesize that this folded conformation is a state related to IFT-A in the retrograde train or it represents a state at the ciliary tip before the assembly of retrograde IFT trains (Fig. 6). Future high-resolution *in situ* structural studies are needed to answer these questions and to unveil the transportation mechanisms of both anterograde and retrograde IFT trains.” (Page 17, lines 343-351)

3. Another novel finding, due to improvement of resolution, is conformation of IFT43

in the base module of IFT-A. This reviewer would recommend to show the detailed density map of IFT43 on other components to prove the loop is visible in the single particle cryo-EM analysis. In Hesketh *et al.* IFT43 was visible only at a short fragment and they employed AlphaFold-multimer to model the rest of IFT43 (Fig.2B). But this modeling is different from experimentally solved structure in this work. Was conformation predicted by AlphaFold2 wrong?

Recently, Hesketh *et al* and Meleppattu *et al* respectively reported the single particle cryo-EM IFT-A structures from human and *Leishmania*. In both structures only the C-terminal region of IFT43 (IFT43_C) is observed^{2,3}. By contrast, our cryo-EM density map allowed us to unambiguously trace almost the entire IFT43 polypeptide chain except for a short disordered loop 50-72 (including both IFT43_N and IFT43_C) (Reviewer Fig. 1). We have emphasized this difference in the revised manuscript as follows.

“Notably, IFT43_N was not observed in recently reported IFT-A structures²⁷⁻³⁰.” (Page 9, lines 164-165)

In Hesketh *et al*, AlphaFold2 was employed to predict the structure of IFT43, in which the predicted IFT43_C (but not IFT43_N) is similar to that in the experimentally solved structures (Reviewer Fig. 2). It should be noted that AlphaFold2 is based on co-evolution information without consideration of physical factor. Therefore, the part of IFT43_N in human and *Leishmania*, lacking interaction information and a compact ternary structure, could not be accurately predicted and modelled by AlphaFold2.

Following this reviewer’s suggestion, we have included the detailed density maps of IFT43 in the revised Supplementary Figs. 5 and 8 (Reviewer Fig. 1).

Reviewer Fig. 1: Density map of IFT43. a (revised Supplementary Fig. 8), IFT43 in IFT-A base module. Atomic models of IFT43_N and IFT43_C are colored in violet and purple,

respectively, while other components and the density map of the base module in grey. **b** (revised Supplementary Fig. 5), Density maps of IFT43 with the atomic model shown in cartoon with sidechains.

Reviewer Fig. 2: Structural comparison of reported IFT43 by us and others. a, Sequence alignment of IFT43. Tt, Hs, Mm, Dm, Cr and Lt are the abbreviations of *Tetrahymena thermophila*, *Homo sapiens*, *Mus musculus*, *Drosophila melanogaster*, *Chlamydomonas reinhardtii* and *Leishmania tarentolae*, respectively. Accession numbers used for IFT43 are Q22NF5 (Tt), Q96FT9 (Hs), Q9DA69 (Mm), Q9VK67 (Dm), A8HYP5 (Cr) and A0A640KKJ7 (Lt). **b**, Structural comparison between IFT43 from *Tetrahymena*, human (pdb-8BBE) and *Leishmania* (pdb-8F50).

Minor points:

4. Fig.1A: It should be shown which part of the sequence was solved in this structural study.

We modified Figure 1a (Reviewer Fig. 3) and the figure legend to label the regions whose structures were solved in our study.

Reviewer Fig. 3 (revised Fig. 1a): Domain organizations of the IFT-A components. IFT144, IFT140, IFT122_N, IFT122_C, IFT121, IFT139 and IFT43 are colored goldenrod, salmon, green, green yellow, turquoise, blue and violet, respectively. Black asterisks indicate the helices inserted between the second and third TPR repeats in IFT144, IFT140,

IFT122, and IFT121. The head and base module components have light yellow and pale blue background colors, respectively. Models of IFT144_{WD1}, IFT144_C, IFT140_C and IFT139_N were built with the aid of AlphaFold-2 and shown in dashed ovals and circles. The middle disordered loop of IFT43 is shown in a dashed line.

5. Show validation map (PAE plot) of AlphaFold2.

We have included the validation maps of AlphaFold-2-predicted IFT139_N, IFT140_C, IFT144_{WD1} and IFT144_C in revised Supplementary Fig. 4c (Reviewer Fig.4).

Reviewer Fig. 4 (revised Supplementary Fig. 4c): Predicted aligned error (PAE) plots of IFT139_N, IFT140_C, IFT144_{WD1} and IFT144_C.

6. Line118, 121, 131: it is not clear how Supplementary Fig. 5 is cited as a reference for pseudo-symmetry of IFT140/144 and 122/121.

Sorry for this confusion. We made a mistake of citing Supplementary Fig. 5 in these positions in the original manuscript. We have corrected it in the revised manuscript.

7. Extended data figure 3: caption is too simple. Each panel must be identified.

We modified Supplementary Fig. 3 (Reviewer Fig. 5) and the figure legend as follows.

Reviewer Fig. 5 (revised Supplementary Fig. 3): Local resolution estimation of IFT-A. **a-g**, Local resolution maps and the corresponding Fourier Shell Correlation curves for IFT-A in elongated state (a), IFT-A in folded state (b), base module at 3.6 Å (c), base module at 4.7 Å (d), head module at 4.2 Å (e), head module at 4.6 Å (f) and head module at 6.0 Å (g).

Reviewer #2:

In this paper, the authors obtained the IFT-A structure from *Tetrahymena*. They did affinity pulldown purification and then cryo-EM to obtain the structures at different conformation from 3.6 - 9 Angstrom resolution. Using AlphaFold and modelling, they presented the model of the complete IFT-A complex and analysed the important interaction in the complex in regard to known mutations affecting IFT structure and function. They also compared their structure with the known in situ structure of *Chlamydomonas* IFT to show the conformation change in the IFT-A.

As someone in the field, I think this is a tremendous work from the authors to obtain the structure, which is elusive for a long time. The paper is presented very nicely in terms of figures and writing. My major concern is that the authors need to show how they modelled better because the resolution of the map is not easy to modelling. In addition, with the limitation of the resolution, the author must justify that the interpretation of the interaction at the amino acid side chain level throughout the paper is appropriate.

NOTE: There are 4 papers presenting the structure of IFT-A recently. I don't think they diminish the work of the authors but it is reasonable to mention or compare them with the structure from the authors.

Thanks. Following this reviewer's suggestion, we included the four related works in the revised text and compared recent IFT-A structures with ours as follows.

“Recently, several studies reported high-resolution cryo-electron microscopy (cryo-EM) IFT-A structures as well as a 20.7-Å *in-situ* cryo-ET model of the anterograde IFT train²⁷⁻³⁰.” (Page 4, lines 60-62)

“Analysis of these structures reveal that the apo IFT-A complex is composed of two rigid structural modules, the head and base modules, and presents two distinct—elongated and folded states in solutions, the latter of which was not identified in recent IFT-A structural studies²⁷⁻³¹.” (Page 4, lines 64-67)

“Notably, IFT43_N was not observed in recently reported IFT-A structures²⁷⁻³⁰.” (Page 9, lines 164-165)

“By contrast, only the elongated conformation with local fluctuations was identified in other IFT-A structures reported recently²⁷⁻³⁰.” (Page 11, lines 205-207)

“In summary, high-resolution atomic models of IFT-A presented here and by others reveal highly similar elongated conformations, providing valuable insights into the assembly of anterograde IFT trains²⁷⁻³⁰. A unique finding of our study is the identification of the folded state of IFT-A that is clearly distinct from those of both

elongated and anterograde-train assembled IFT-A conformations. We hypothesize that this folded conformation is a state related to IFT-A in the retrograde train or it represents a state at the ciliary tip before the assembly of retrograde IFT trains (Fig. 6). Future high-resolution *in situ* structural studies are needed to answer these questions and to unveil the transportation mechanisms of both anterograde and retrograde IFT trains.” (Page 17, lines 343-351)

My comments are as follows:

1. While the authors claim 3.6, 4.2 and 4.6 in some map, the map doesn't show the feature very well, especially the 4.6 Angstrom map, the helix are not very well separated, perhaps due to the threshold being too low for rendering. Is this possible to show the model well fitted in the map to validate the feature there and probably the side chain fitting visualization in the 3.6 Angstrom map. Somehow in Figure S4 but a bit better highlighting.

Model building of IFT-A in this study is performed as follows.

1. Based on the density map of the base module at 3.6 Å, we built the atomic models of IFT121, IFT139_C (residues 695-1387), IFT122_N (residues 1-705) and IFT43. Similarly, we obtained the atomic models of IFT122_C (residues 706-1251), IFT140_N (residues 1-1081), IFT144 (residues 351-1129) using the density map of head module at 4.2 Å. Continuity of the two density maps allows to unambiguously trace these fragments.
2. Due to the flexibility, models of IFT139_N (residues 1-694) of the base module, IFT144_{WD1} and IFT144_C (residues 1130-1387) of the head module were built with the aid of AlphaFold-2.
3. Since IFT140_C (residues 1082-1407) is only visible in the map of the head module at 4.6 Å, model of IFT140_C (residues 1082-1407) was built with the assistance of AlphaFold-2 in this density map.

Following this reviewer's suggestion, we modified Supplementary Fig. 4 (Reviewer Fig. 6) and also provided representative density maps of components of the base module at 3.6 Å and the head module at 4.2 Å resolutions in revised Supplementary Fig. 5 (Reviewer Fig. 7).

Reviewer Fig. 6 (revised Supplementary Fig. 4): Cryo-EM density maps of the IFT-A complex. **a**, High-resolution density maps of the head and base modules in two orthogonal views. The IFT-A components are colored and the color scheme is shown on the right. **b**, Cryo-EM density maps of the IFT-A subunits with the atomic models in cartoon representation. **c**, Predicted aligned error (PAE) plots of AlphaFold-2 predicted IFT139_n, IFT140_c, IFT144_{wd1} and IFT144_c.

Reviewer Fig. 7 (revised Supplementary Fig. 5): Representative cryo-EM density maps for the selected elements of each component of IFT-A.

2. With that resolution, how did the author distinguish IFT144 and IFT140 or IFT121 from IFT122 for modelling?

In order to manually build models of IFT-A and distinguish two components with similar domain organization (such as IFT144 and IFT140, IFT121 and IFT122), we focused on the unique residues with large sidechains at the beginning and end of each domain for component assignment. Aromatic residues at the beginning (residues 1-10) and the end (residues 331-340) of IFT121_{WD1} allowed us to distinguish IFT121 from IFT122 (Reviewer Fig. 7). Similarly, the last β -sheet of IFT140_{WD2} (residues 705-714) allowed us to distinguish IFT140 from IFT144 (Reviewer Fig. 7). Moreover, the continuity of the density maps of the base module at 3.6 Å and the head module at 4.2 Å allowed us to unambiguously trace majority of the complex. We have revised model building part in the method section as follows.

“We combined *de-novo* model building and rigid-body docking to generate the atomic models of the IFT-A complex. In order to manually build models of IFT-A in Coot, we

focused on the unique residues with large sidechains at the beginning and end of each domain for component assignment. Aromatic residues at the beginning (residues 1-10) and the end (residues 331-340) of IFT121_{WD1} allowed us to distinguish IFT121 from IFT122. Similarly, the last β -sheet of IFT140_{WD2} (residues 705-714) allowed us to distinguish IFT140 from IFT144. The continuity of density maps of the base module at 3.6 Å and the head module at 4.2 Å allowed us to unambiguously trace most of the IFT-A residues. Due to the flexibility, AlphaFold-2 was employed to assist model building for IFT139_N (residues 1-694) of the base module, IFT144_{WD1}, IFT144_C (residues 1130-1387) and IFT140_C (residues 1082-1407) of the head module³². (Page 23, lines 445-455)

3. Also, with the modelling at 4.2 and 4.6 resolution, how does the author justify their interpretation of side chain interaction like in Figure 2f or Fig. S5c.

In Fig. 2f and Supplementary Fig. 5c, we analyzed the inter-molecular interactions (IFT43-IFT122-IFT121 and IFT139-IFT122) in the base module at 3.6 Å. Local density maps involved are shown in revised Supplementary Fig.5 (Reviewer Fig. 8), which allowed the interpretation of side-chain interactions in these regions.

Reviewer Fig. 8 (from revised Supplementary Fig. 5): Representative cryo-EM density maps for the selected elements of IFT43 and IFT139.

4. Fig. 4a Opening of IFT140_{TPR} is not clear from 3 transition figures.

We have modified Fig. 4a (Reviewer Fig. 9) to clearly show the opening of IFT140_{TPR} during the assembly of IFT-A into the anterograde train.

Reviewer Fig. 9 (revised Fig. 4a): Fitting of IFT-A into the cryo-ET density map of the anterograde train. The head and base cores are colored in light yellow and pale blue as Fig. 1e, and the flexible TPRs in the elongated and assembled states colored in pink and orange, respectively.

5. It is important to label directly in Figure which one is not from this paper such as in Extended Figure 10.

Corrected.

Reviewer #3:

The manuscript by Ma et al provides important structural insights into the intraflagellar transport complex A (IFT-A) and its assembly in the anterograde IFT trains. This is a well written and thorough study describing novel conformations of IFT-A and provides detailed insights into how IFT-A monomers could incorporate into trains. The likely biological significance of the folded conformation in retrograde IFT, although speculative, is another highlight of the paper. Considering other papers in the field being published, this paper should be published immediately without any delay.

My comments below are mostly editorial and should be considered by the authors as they feel necessary.

1. Fig 2: IFT144 and IFT140 colors are too similar and could be changed.

Thanks for this good point. We have modified the coloring scheme in all figures in the revised manuscript. Revised Fig. 2 (Reviewer Fig. 10) is shown below.

Reviewer Fig. 10 (revised Fig. 2): The head and base modules of IFT-A. a-b, Atomic models of the head (a) and base (b) modules. The color scheme is shown. The cartoon indicates the relative position of the modules in the elongated state of IFT-A. c-d, The pseudo-two-fold symmetric IFT144-IFT140 (c) and IFT122-IFT121 heterodimers (d). The central TPR junctions formed by the first four TPRs are highlighted in black dashed boxes, which are enlarged on the right. The TPRs are numbered and the inserted α -helix in each subunit is labeled with an asterisk. e, A different view of the head module with IFT122

shown in the surface representation. **f**, Close-up views of the interaction network in the base module between IFT122_{WD1} and IFT139_C (1), between IFT121_{WD1} and IFT43 (2), between IFT121_{WD2} and IFT43 (3) and between IFT43 and IFT139-IFT121-IFT122 (4).

2. Fig. 4. IFT-A and B interactions, including that between IFT74 and Thm1(IFT139) can be better highlighted in text in the results and figures. The authors might consider including details from Fig S10 in Fig 4c.

Following this reviewer's suggestion, we revised both Fig. 4c (Reviewer Fig. 11) and Supplementary Fig. 11.

Reviewer Fig. 11 (revised Fig. 4c): Fitting of atomic models of *Tetrahymena* IFT-A and *Chlamydomonas* IFT-B into the anterograde train map unveiling the putative [IFT-A]-[IFT-B] interfaces. IFT-A is from this study and colored as Fig. 1a, and IFT-B is from Lacey *et al.* with IFT172 (residues 1-1104) colored in cyan, IFT88 (residues 122-690) in yellow, IFT74 (residues 135-340) in orange and the rest of IFT-B in light brown. EMD code of microtubule doublet (MTD) are EMD-20631. PDB codes of IFT-B, dynein-2 motor domain and tail domain are 8BD7, 6RLA and 6RLB, respectively.

We revised the text in the result and discussion sections as follows.

“The interior surface on the other side of IFT-A is characterized by convex-shaped TPR clusters from IFT140, IFT144 and IFT139 (Fig. 4c and Supplementary Fig. 11). To gain more insights into the interactions between IFT-A and IFT-B in the anterograde train, we built a pseudo-atomic anterograde train model by docking our assembled IFT-A structure and the published IFT-B structure into a composite map of the anterograde IFT train (Fig. 4c)²⁷. The model unveils three putative contacting areas between IFT-A and IFT-B, IFT140^{IFT-A}-IFT172^{IFT-B}, IFT144^{IFT-A}-IFT88^{IFT-B}, and IFT139^{IFT-A}-IFT74/81^{IFT-B} (Fig. 4c), which are consistent with previous biochemical studies^{15,27,29,44-47}. This assembled configuration of IFT-A in the anterograde train is in line with the notion that the structural unit of the anterograde trains is made of IFT-A, IFT-B and dynein-1b in a ratio of 3:6:2 (Fig. 4d)¹⁵.” (Page 13, lines 249-259)

“It was reported that IFT74^{IFT-B} and IFT139^{IFT-A} are required for the association between IFT-A and IFT-B^{15,47}. Concordantly, model fitting of our high-resolution IFT-A structure into the *in situ* density map of the anterograde train from green alga *Chlamydomonas* unveils that IFT74^{IFT-B} is in close vicinity to IFT139^{IFT-A} (Fig. 4c and Supplementary Fig. 11)^{15,27}.” (Page 16, lines 310-314)

3. Fig. 4. IFT-A and B interactions, where do the authors think that the IFT140 interaction is happening with IFT-B? Based on Fig S10, the authors marked IFT172 close to IFT140, and if so, the authors should probably state it in the text.

Following this reviewer’s suggestion, we revised the text in the result section in the revised manuscript.

“The interior surface on the other side of IFT-A is characterized by convex-shaped TPR clusters from IFT140, IFT144 and IFT139 (Fig. 4c and Supplementary Fig. 11). To gain more insights into the interactions between IFT-A and IFT-B in the anterograde train, we built a pseudo-atomic anterograde train model by docking our assembled IFT-A structure and the published IFT-B structure into a composite map of the anterograde IFT train (Fig. 4c)²⁷. The model unveils three putative contacting areas between IFT-A and IFT-B, IFT140^{IFT-A}-IFT172^{IFT-B}, IFT144^{IFT-A}-IFT88^{IFT-B}, and IFT139^{IFT-A}-IFT74/81^{IFT-B} (Fig. 4c), which are consistent with previous biochemical studies^{15,27,29,44-47}. This assembled configuration of IFT-A in the anterograde train is in line with the notion that the structural unit of the anterograde trains is made of IFT-A, IFT-B and dynein-1b in a ratio of 3:6:2 (Fig. 4d)¹⁵.” (Page 13, lines 249-259)

“In addition to the IFT139^{IFT-A}-IFT74^{IFT-B} contact at the IFT-A base module, IFT-A fitting into the anterograde train also reveals spatial proximities between IFT88^{IFT-B} and IFT144^{IFT-A} and between IFT172^{IFT-B} and IFT140^{IFT-A} (Fig. 4c and Supplementary Fig. 11)^{29,44-46}.” (Page 16, lines 318-321)

4. Fig. 4. IFT-A and B interactions, where do the authors think that the IFT144 interaction is happening with IFT-B? Please comment in text if the authors think that IFT88 is involved or not.

Please refer to the answer to Comment #3.

5. Extended Fig 9. This figure is very speculative, but again is a nice addition to the paper and could be included in the main figures. The IFT139 regions interacting with IFT74 could be highlighted in text and figures, in line with the description of the type III mutations. Same for type III mutations described in IFT144 and IFT140, if any.

Follow this reviewer’s suggestion, we revised and moved Supplementary Fig. 9 to Fig. 5. Possible type II and III mutations of IFT-A interrupting the [IFT-A]-[IFT-A] and [IFT-

A]-[IFT-B] interactions are enlarged and included in reviewer Fig. 12 (revised Supplementary Fig. 12) for clarity.

Reviewer Fig. 12 (revised Supplementary Fig. 12): Enlarged views of mutations at type-II and -III interfaces. **a**, Human disease mutations are mapped onto the assembled IFT-A of anterograde trains. Dashed boxes in colors indicate type-II mutations and closeup views of those mutations at the interfaces are shown below. **b**, The [IFT-A]-[IFT-B] interfaces. Closeup views of the type-III mutations at the [IFT-A]-[IFT-B] interfaces are shown below. Mutations of type I-IV are shown in blue, red, black and purple dots, respectively.

We revised the text in the result section as follows.

“Notably, all type-III mutants are located either in IFT144_C or in IFT139_N, in line with the observation that both IFT144 and IFT 139 are at the interface between IFT-A and IFT-B (Supplementary Fig. 12). Intriguingly, human IFT139, in spite of its periphery position in IFT-A and lacking WDs, is a frequently disease-causing subunits among all IFT-A proteins (Fig. 5, Supplementary Figs. 6,12 and Supplementary Data 1).” (Page 14, lines 278-283)

6. Lines 304-5: The results showing IFTA-B subunit interactions should be introduced in this part of the Results section for clarity.

We revised the text in the result and discussion sections to include the [IFT-A]-[IFT-B] interactions.

“The interior surface on the other side of IFT-A is characterized by convex-shaped TPR

clusters from IFT140, IFT144 and IFT139 (Fig. 4c and Supplementary Fig. 11). To gain more insights into the interactions between IFT-A and IFT-B in the anterograde train, we built a pseudo-atomic anterograde train model by docking our assembled IFT-A structure and the published IFT-B structure into a composite map of the anterograde IFT train (Fig. 4c)²⁷. The model unveils three putative contacting areas between IFT-A and IFT-B, IFT140^{IFT-A}-IFT172^{IFT-B}, IFT144^{IFT-A}-IFT88^{IFT-B}, and IFT139^{IFT-A}-IFT74/81^{IFT-B} (Fig. 4c), which are consistent with previous biochemical studies^{15,27,29,44-47}. This assembled configuration of IFT-A in the anterograde train is in line with the notion that the structural unit of the anterograde trains is made of IFT-A, IFT-B and dynein-1b in a ratio of 3:6:2 (Fig. 4d)¹⁵.” (Page 13, lines 249-259)

“It was reported that IFT74^{IFT-B} and IFT139^{IFT-A} are required for the association between IFT-A and IFT-B^{15,47}. Concordantly, model fitting of our high-resolution IFT-A structure into the *in situ* density map of the anterograde train from green alga *Chlamydomonas* unveils that IFT74^{IFT-B} is in close vicinity to IFT139^{IFT-A} (Fig. 4c and Supplementary Fig. 11)^{15,27}.” (Page 16, lines 310-314)

“In addition to the IFT139^{IFT-A}-IFT74^{IFT-B} contact at the IFT-A base module, IFT-A fitting into the anterograde train also reveals spatial proximities between IFT88^{IFT-B} and IFT144^{IFT-A} and between IFT172^{IFT-B} and IFT140^{IFT-A} (Fig. 4c and Supplementary Fig. 11)^{29,44-46}.” (Page 16, lines 318-321)

7. Lines 215-217: The sentence can be simplified.

We simplified this sentence in the revised manuscript as follows.

“Calculation of the relative orientation between the head and base modules of IFT-A in the anterograde IFT train clearly unveiled that IFT-A in the anterograde train is more similar to the elongated state than to the folded state.” (Page 11, lines 218-220)

8. Line 164: the authors might consider showing the seven IFT43 helices. I don't see them currently shown clearly in Extended Fig. 4.

We revised Supplementary Fig. 4 (reviewer Fig. 6) to clearly show the seven IFT43 helices.

References

1. Lacey, S.E., Foster, H.E. & Pigino, G. The molecular structure of IFT-A and IFT-B in anterograde intraflagellar transport trains. *Nat Struct Mol Biol* (2023).
2. Meleppattu, S., Zhou, H., Dai, J., Gui, M. & Brown, A. Mechanism of IFT-A polymerization into trains for ciliary transport. *Cell* **185**, 4986-4998 e12 (2022).
3. Hesketh, S.J., Mukhopadhyay, A.G., Nakamura, D., Toropova, K. & Roberts, A.J. IFT-A structure reveals carriages for membrane protein transport into cilia. *Cell* **185**, 4971-4985 e16

(2022).

4. Jiang, M. et al. Human IFT-A complex structures provide molecular insights into ciliary transport. *Cell Res* (2023).